# Spontaneous Transition of Alkyl Carbocations to Unsaturated Vinyl-Type Carbocations in Organic Solutions

**DOI:** 10.3390/ijms24021802

**Published:** 2023-01-16

**Authors:** Evgenii S. Stoyanov, Irina V. Stoyanova

**Affiliations:** N.N. Vorozhtsov Institute of Organic Chemistry, Siberian Branch of Russian Academy of Sciences, 630090 Novosibirsk, Russia

**Keywords:** vinyl and alkyl carbocations, carborane salts, IR spectroscopy

## Abstract

It was found that alkyl carbocations, when their salts are dissolved in common organochlorine solvents, decompose to unsaturated vinyl-type carbocations that are stabler in solutions. This is a convenient method for obtaining salts of vinyl cations and their solutions for further research.

## 1. Introduction

To date, alkyl carbocations have been investigated in solutions of only liquid superacids by nuclear magnetic resonance (NMR) and infrared (IR) spectroscopy [1,2]. There are no reports on their salts in solutions of organic solvents common in chemical practice, except for our work [3] in which we present H^1^ NMR and IR spectra of the t-Bu^+^ cation in solutions of its salt in d_4_-dichloroethane (DCE) and d_2_-dichloromethane (DCM). It turned out that in such solutions, alkyl carbocations are unstable and converted to other carbocations.

Our knowledge about the simplest alkene carbocations not stabilized by heteroatoms or aryl groups remains more limited, despite extensive theoretical and experimental research in the past five decades [4,5,6,7,8,9,10,11,12]. From studies on solvolysis reactions, it has been concluded that vinyl-type carbocations are highly reactive and therefore difficult to study [5,7,10]. On the other hand, their expected reactivity is likely exaggerated, as demonstrated by Mayr and coworkers [13]. Recently, the high reactivity of the benzyl carbocation was explained: when acting as a strong protonating agent, it converts into a carbene molecule, whose high reactivity can be perceived as the reactivity of the carbocation [14]. The same conclusions have been reached by M. Niggemann and S. Gao [15]: the high reactivity of vinyl cations is a myth; they behave like reactive intermediates with carbene-like reactivity.

Numerous attempts have been made to study the allyl cation by NMR spectroscopy in liquid superacids at low temperatures, and they have failed [16]. The formation of C_3_H_5_^+^ has been proved in a cryogenic super-acidic matrix (170 K) by IR spectroscopy [17,18]. With the temperature increasing up to 230 K, the IR spectrum of the sample changes, from which it was concluded that non-stabilized alkene carbocations are stable only at low temperatures (below −100 °C). However, this turned out to be incorrect. Recently, vinyl-type carbocations C_3_H_5_^+^ and C_4_H_7_^+^ were obtained as carborane salts, which are thermally stable up to 150 °C; their crystals were grown from salt solutions in dichloromethane (DCM) and were characterized by X-ray crystallography and IR spectroscopy [19,20,21]. That is, salts of vinyl carbocations are stable in DCM solutions at room temperature, and solvolysis reactions do not occur.

In the present work, we report the results of an IR spectroscopic analysis of alkyl carbocations’ transformation (using t-Bu^+^ and methyl-cyclopentyl^+^ as an example) into vinyl-type carbocations (X-ray crystallographic data were used as evidence) [19,20,21] in solutions in weakly basic organochlorine solvents commonly used in chemical practice. As a counterion, the undecachlorocarborane anion, CHB_11_Cl_11_^−^, was chosen (hereinafter abbreviated as {Cl_11_^−^}, Appendix A) because of its extreme stability and very low basicity, which promotes the formation of stable ionic salts with highly acidic cations [22]. Salts t-Bu^+^{Cl_11_^−^} and methylcyclopentyl^+^{Cl_11_^−^} were prepared as described in [3,23].

## 2. Results and Discussion

The IR spectrum of a solution of the t-Bu^+^{Cl_11_^−^} salt in d_4_-DCE shows a strong absorption pattern of the t-Bu^+^ cation, and this absorption matches that of crystal salt t-Bu^+^{Cl_11_^−^}, characterized by X-ray crystallography [3] (Figure 1(a), red and black, respectively). There is also clear-cut weak absorption with a characteristic band at 1534 cm^−1^ from another compound, which we will denote as X. Its intensity strongly increases in the spectrum of a solution of the t-Bu^+^{Cl_11_^−^} in CD_2_Cl_2_, simultaneously with the appearance of the second C=C stretch band at 1490 cm^−1^, while absorption intensity of the t-Bu^+^ cation decreases correspondingly (Figure 1(b), red). The spectrum of the t-Bu^+^ cation is actually the same between the solutions in the two solvents. Therefore, it is possible to subtract the spectrum of a solute in d_4_-DCE with strong absorption of t-Bu^+^ (Figure 1(a), red) from that of a solution in d_2_-DCM with weak absorption of t-Bu^+^ (Figure 1(b), red) up to its full compensation (both spectra are subjected to solvent absorption compensation). This approach made it possible to isolate compound X’s spectrum of acceptable quality (Figure 1(b), green). At the same time, an intense spectrum of the anion {Cl_11_^−^} is preserved, which means that X is a cation.

The solubility of the t-Bu^+^{Cl_11_^−^} salt in 1,2-dibromoethane is low. The weak IR spectrum of the solute contains representative frequencies 1535 and 1490 cm^−1^ of the bands of cation X of the dissolved salt (Figure 1(c), red). Evaporation of a drop of the solution on the surface of the crystal of an ATR accessory allowed observation of the spectrum of the X^+^{Cl_11_^−^} salt in more detail, without residual bands from the solvent subtraction (Figure 1(c), blue). The solubility of the t-Bu^+^{Cl_11_^−^} salt is greater in d_2_-1,1,2,2,-tetrachloroethane (TCE), enabling us to record a higher-quality spectrum of the dissolved salt (Figure 1(d), red). The spectrum contains no absorption from the t-Bu^+^ cation but only from the X^+^ cation. The spectrum of the dry residue from the evaporated solution matches that of the dissolved salt (Figure 1(d), blue).

The IR spectrum of the X^+^ cation shows two bands: at ~1490 and 1535 cm^−1^, known as characteristic frequencies of C=C stretching vibrations of the isobutylene cation [19]. The former corresponds to this cation in a homogeneous environment of solvent molecules in solutions or of {Cl_11_^−^} anions in the crystalline salt. The latter frequency matches the isobutylene cation involved in ion pairing with anion {Cl_11_^−^} in solutions and in a solid phase. Therefore, cation X^+^ is expected to be the isobutylene carbocation, (CH_3_)_3_C=CH^+^. Strict proof is the finding that from solutions of t-Bu^+^{Cl_11_^−^} in DCM and TCE, crystals formed successfully, and their X-ray diffraction analysis showed that these are the previously studied crystals of salt (CH_3_)_3_C=CH^+^{Cl_11_^−^} (which were originally obtained from solutions of C_4_H_7_^+^{Cl_11_^−^} in DCM) [19].

Thus, when the t-Bu^+^{Cl_11_^−^} salt is dissolved in organochlorine solvents, it dissociates with the formation of solvent-separated ion pairs, ref. [3] and solvated cation t-Bu^+^_solv_ decomposes into the isobutylene cation:(1)$$t-{\mathrm{Bu}}^{+}\{{\mathrm{Cl}}_{11}^{-}\}\overset{\mathrm{solvent}}{\to }t-{\mathrm{Bu}}_{\mathrm{solv}}^{+}\{{\mathrm{Cl}}_{11}^{-}\}\underset{}{\to }{\left({\mathrm{CH}}_{3}\right)}_{2}C={\mathrm{CH}}_{\mathrm{solv}}^{+}\{{\mathrm{Cl}}_{11}^{-}\}+H_{2}$$

The course of reaction 1 is confirmed by our finding that the t-Bu^+^ cation salt dissolves slowly with active stirring. The spectra of solutions presented in Figure 1 belong to freshly prepared solutions (within 1–2 min) that are in equilibrium with a powder of t-Bu^+^{Cl_11_^−^}. If the dissolution is carried out for a long time (hours), then solid t-Bu^+^{Cl_11_^−^} can be dissolved completely. In the spectrum of such a solution, the absorption intensity of the (CH_3_)_3_C=CH^+^ cation increases significantly, while the absorption of the t-Bu^+^ cation may not be detectable. Thus, the spectra of cations (CH_3_)_3_C=CH^+^ and t-Bu^+^ shown in Figure 1 are time and concentration dependent, because the transition of the t-Bu^+^ cation to the butylene cation requires a rather long time, and the solubility of the C_4_H_7_^+^{Cl_11_^−^} salt is higher than that of t-Bu^+^{Cl_11_^−^}. 

Characterization of the H_2_ formed according to Equation (1) is a difficult task: in the NMR spectrum of a solution prepared in a sealed NMR tube, in the expected region of the NMR ^1^H signal from H_2_ dissolved in DCM (4.152 ppm), there are other weak signals from impurities, which make the identification unreliable.

Dissolution of another salt of alkyl carbocation, methyl-cyclopentyl^+^ [CH_3_-C_5_H_8_^+^{Cl_11_^−^}], in the same solvents, leads to similar but more complex changes in the IR spectrum of the cation. The similarity lies in the finding that the weakened absorption pattern of the initial methyl-cyclopentyl^+^ (its IR spectrum was interpreted in [24]) can remain in the spectra of solutions or disappear altogether. Instead, the absorption of an unsaturated carbocation appears with characteristic bands of the C=C stretching [21] in the frequency range 1460−1535 cm^−1^. Meanwhile, the spectra are dependent both on the duration of the salt dissolution and on its concentration. The distinction is that two bands of the C=C^+^ stretching appear at frequencies of 1460 and 1490 cm^−1^, which are typical for cations in homogeneous immediate surroundings without involvement in the formation of ion pairs [21]. This means that two different carbocations are formed, which we will denote by Y (having the 1460 cm^−1^ band) and Z (having the 1490 cm^−1^ band). A pure spectrum of the Y cation, not contaminated by side products, was obtained in a freshly prepared solution of CH_3_-C_5_H_8_^+^{Cl_11_^−^} in DCM (Figure 2). Over time, the spectrum of this solution changes: νC=C band intensity at 1460 cm^−1^ decreases but emerges and increases for the pair of C=C^+^ stretch bands at 1563 and 1490 cm^−1^ of the Z cation (Figure 2 and Appendix A). They correspond to those of vinyl cations in contact (with anion {Cl_11_^−^}) and solvate-separated ion pairs, respectively [21]. After 1–2 days, crystals grow from this solution. X-ray diffraction analysis indicated that this is a previously studied salt (CH_3_−C^+^=CH−CH_3_){Cl_11_^−^} [19], having one band of νC=C in the IR spectrum at 1490 cm^−1^. Thus, the Z^+^ cation is CH_3_−C^+^=CH−CH_3_.

We failed to obtain salt crystals of the Y^+^ cation; therefore, it had to be identified based on its IR spectra. The spectrum of cation Y^+^ in solutions contains one intense νC=C band at 1460 cm^−1^. It is lower in frequency than that of vinyl-type cations C_4_H_7_^+^ and C_3_H_5_^+^, in which a substantial localization of a positive charge on the C=C bond ensures its lowest vibrational frequency at 1490 cm^−1^ [20,21]. Only the asymmetric 
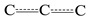
 stretch vibration of allyl cation (CH_2_CHCH_2_)^+^ has a lower frequency [20], at 1303 cm^−1^. Therefore, it can be assumed that the Y^+^ cation is of the allyl type. It forms from the alkyl cation of methyl-cyclopentyl^+^ in a solvent medium and, being unstable, subsequently decomposes into butylene cation Z and other products (Figure 1).

The following findings support the notion that allyl cation Y^+^ is formed: (i) at the first stage, the methyl-cyclopentyl^+^ cation decomposes only into one unsaturated carbocation without the formation of other accompanying carbon-containing products; therefore, it can be a cation of composition C_6_H_9_^+^; (ii) only the asymmetric vibration of the 
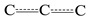
 group with close-to-aromatic CC bonds—on which the positive charge is predominantly concentrated—can have a frequency lower than that of the C=C^+^ stretch; (iii) the CH_3_ group of the cation, in contrast to methylcyclopentyl^+^ [23], does not participate in hyperconjugation because it is bonded to the sp^2^ C atom with a field p_z_ orbital. The Y^+^ cation is unstable and decomposes yielding two or more unsaturated carbocations. One of them is the butylene cation, the salt of which crystallizes out and allows it to be uniquely identified by X-ray diffraction.

From IR spectra of freshly prepared solutions of CH_3_-C_5_H_8_^+^{Cl_11_^−^} in other solvents, d_4_-DCE and d_2_-1,1,2,2- TCE, it follows that both cations Y^+^ and Z^+^ are formed in them in different ratios, depending on the nature of the solvent and solute concentration (Figure 2, bands of C=C stretching at ~1490 and 1457/1460 cm^−1^).

The finding that during the spontaneous decomposition of the methyl-*cyclo*-pentyl^+^ cation in solutions one of the resulting products is the butylene^+^ cation is surprising. Repeated reproduction of the experiment on crystals grown from solutions of CH_3_-C_5_H_8_^+^{Cl_11_^−^} (either via slow evaporation of the solvent or by keeping the solution over CCl_4_ vapor) always gave rise to salt crystals (CH_3_−C^+^=CH−CH_3_){Cl_11_^−^} exclusively, surrounded by a waxy product or dark non-crystalline powder of variable composition. The waxy by-products forming along the way include neutral hydrocarbons, which can be extracted with CCl_4_ and identified by IR or NMR spectroscopy, and other carbocations with C=C stretch frequencies in the range of 1480–1580 cm^−1^. Thus, the rupture of the CC bond in vinyl cations was established experimentally. It is of interest and requires a separate study.

The transition from an alkyl to alkene carbocations is also documented for the propyl^+^ cation in salt C_3_H_5_^+^{Cl_11_^−^}. The saturated chloronium cation, *cyclo*-(CH_2_)_4_Cl^+^, manifests the same property when its salt with the {Cl_11_^−^} anion is dissolved in DCM: it spontaneously transforms into vinyl cations with cycle opening in two ways with the release of H_2_ or HCl [21]. It seems that this property is common to all small alkyl carbocations.

This work opens up opportunities for NMR research on vinyl cations and their transformations in solutions. Caution is advised because NMR spectra of vinyl carbocations in organochlorine solvents have their own peculiarities, which must be studied and explained before this method is used to detect the formation of these cations in solutions. (The main peculiarity is the broadening of the NMR signals, caused by solvation and self-association of ion pairs, which leads to the disappearance of the fine structure of the NMR signals. We succeeded in obtaining a high-quality ^1^H and ^13^C NMR spectrum of the CHCl=CHCH^+^CH_3_ cation in a solution of its carborane salt in SO_2_ClF: the saturated solution was aged in a sealed NMR tube for 3 days and a portion of the dissolved salt was isolated as a thin coating on the surface of the tube, reducing the concentration of the solute by five times. This procedure eliminated the self-association. The NMR spectrum after a long scan showed a detailed hyperfine structure, which allowed for a confident interpretation [25]).

In conclusion, here we report the discovery of an unusual spontaneous transition of alkyl carbocations to unsaturated ones at room temperature after their salts are dissolved in weakly basic organochlorine solvents. The molecular mechanism of this transition are yet to be studied. At the same time, other important results were obtained: (1) solvated saturated carbocations (C_n_H_2n+1_)_solv_ are less stable than unsaturated ones (C_n_H_2n−1_)_solv_, and this property may facilitate the transition of the former into the latter; (2) vinyl cations are soluble in common organochlorine solvents without interaction with the solvent, i.e., there are no solvolysis reactions; (3) under the action of solvation in solutions, the CC bond of vinyl cations can spontaneously break yielding products with shorter and longer C_n_ chains of carbon atoms. This work offers a simple way to obtain pure crystalline salts of certain stable vinyl carbocations, such as isobutylene^+^ or chain butylene^+^, from cationic alkyne precursors whose synthesis has been developed [3,23]. In the case of the formation of unstable unsaturated carbocations (e.g., from methyl-cyclopentyl^+^) that undergo further changes, it is now possible to characterize their properties by NMR spectroscopy in solvents common in chemical practice.

## 3. Methods and Materials

Experimental details of obtaining the studied samples and the conditions for recording their IR spectra are given in Appendix A.

The English language was corrected and certified by shevchuk-editing.com accessed on 3 January 2023.

## Data Availability

Not applicable.

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
