# Peer review of "Spontaneous Transition of Alkyl Carbocations to Unsaturated Vinyl-Type Carbocations in Organic Solutions"

_ijms, 2023, doi:10.3390/ijms24021802_

Round 1

Reviewer 1 Report

The manuscript “Spontaneous transition of alkyl carbocations to unsaturated vinyl-type carbocations in organic solutions” discuss the behavior of t-Bu+ and methyl-cyclopentyl+ carbocations in chlorinated organic solvents such as dichloromethane, dichloroethane and 1,1,2,2-tetrachloroethane. Superacid H[CHB11Cl11] was used as counterion because it stabilize salts of highly acidic cations.

It was shown that in IR spectra of carbocation salts appear bands of double carbon-carbon bonds i.e. t-Bu cation transformed to isobutylene cation.

In the case of methylcyclopentyl cation which obtained in situ from hexane also transformed to vinyl-type carbocation. Presence of last one was detected by IR too. It is interesting that carbocation named “Y” undergoes further transformations to linear product.

The results obtained are not in doubt. Conclusions fully and correctly reflect the content of the investigation

I believe that the manuscript can be published in present form.

Author Response

Referee 1 has no questions and recommended to publish this ms as it is

Reviewer 2 Report

The topic of the article is very important, nevertheless the way that the authors support their findings is insufficient and for this reason, I do not recommend publication. Specifically, the authors rely specifically on IR spectroscopy for the characterization of the again very interesting chemical species and they completely ignore NMR spectroscopy which is the method of choice for characterizing carbocations (George Olah won the Nobel prize in chemistry by using NMR spectroscopy on his characterization of carbocations). It is without a doubt that the authors may be experts in IR but the technique not coupled with other ones is insufficient and can lead to wrong conclusions. In conclusion, the findings cannot be published in a reputable scientific journal without NMR. In regard to the X-ray characterization that appears in a past publication, it is known that X-ray is not an ideal method for locating the protons.

Other minor issues are that the structure of the paper is not proper, lacking an introduction, experimental, results and discussion section and finally a conclusion. It rather reads as an essay of theories that are only IR supported and in my opinion are wrong.

Author Response

Comments and Suggestions for Authors

The topic of the article is very important, nevertheless the way that the authors support their findings is insufficient and for this reason, I do not recommend publication. Specifically, the authors rely specifically on IR spectroscopy for the characterization of the again very interesting chemical species and they completely ignore NMR spectroscopy which is the method of choice for characterizing carbocations (George Olah won the Nobel prize in chemistry by using NMR spectroscopy on his characterization of carbocations). It is without a doubt that the authors may be experts in IR but the technique not coupled with other ones is insufficient and can lead to wrong conclusions. In conclusion, the findings cannot be published in a reputable scientific journal without NMR. In regard to the X-ray characterization that appears in a past publication, it is known that X-ray is not an ideal method for locating the protons.

Other minor issues are that the structure of the paper is not proper, lacking an introduction, experimental, results and discussion section and finally a conclusion. It rather reads as an essay of theories that are only IR supported and in my opinion are wrong.

Below is a detailed response to the Referee 2 report.

Specifically, the authors rely specifically on IR spectroscopy for the characterization of the again very interesting chemical species.

This is not true. In this work, all evidence for the composition and structure of carbocations is based on X-ray diffraction analysis of single crystals. And these results cannot be questioned. IR spectroscopy is used only to transfer the results of the study of carbocations in crystals to their content in the solutions from which the crystals are grown. This transfer of results is based on the fact that the IR spectra of carbocations in crystals and in solutions coincide. It is impossible to use the NMR method for this: it is not suitable for establishing such a connection between crystals and a solution. To perform this work, you do not need to be a specialist in IR spectroscopy, since the interpretation of the IR spectra of cations was performed by us earlier (all references are given).

....and they completely ignore NMR spectroscopy which is the method of choice for characterizing carbocations (George Olah won the Nobel prize in chemistry by using NMR spectroscopy on his characterization of carbocations).

This is not true at all. Looks like the Referee didn't read the article. It contains the paragraph «This work opens up opportunities for NMR research on vinyl cations and their transformations in solutions. Caution is advised because NMR spectra of vinyl carbocations in organochlorine solvents have their own peculiarities, which must be studied and explained before this method is used to detect the formation of these cations in solutions. (The main peculiarity is the broadening of the NMR signals, caused by solvation and self-association of ion pairs, which leads to the disappearance of the fine structure of the NMR signals. We succeeded in obtaining a high-quality 1H and 13C NMR spectrum of CHCl=CHCH+CH3 cation in a solution of its carborane salt in SO2ClF: the saturated solution was aged in a sealed NMR tube for 3 days and a portion of the dissolved salt was isolated as a thin coating on the surface of the tube, reducing the concentration of the solute by 5 times. This procedure eliminated the self-association. The NMR spectrum after a long scan showed detailed hyperfine structure, which allowed for a confident interpretation [25].) This paragraph explains why NMR cannot be used, and I will explain this in more detail below..

About George Olah. He and co-authors characterized by NMR only alkane and aril carbocations in superacid solutions. He tried for many years to obtain vinyl carbocations, but without success because these cations cannot exist in liquid superacids. We are the first to obtain salts of vinyl cations using solid carborane superacids, obtained by Prof. Christopher Reed, a former employee of George Olah, who left his laboratory and organized his own for the synthesis of solid superacids. I collaborated with Christopher Reed for many years and was the first to obtain vinyl carbocations using his carborane acids. Last year we were the first to publish in IJMS high-resolution 1H/13C NMR spectra of vinyl carbocation and 1H/13C MAS NNR spectra of methyl propargyl carbocation, CHºC-C+H-CH3.

It is without a doubt that the authors may be experts in IR but the technique not coupled with other ones is insufficient and can lead to wrong conclusions.

As I already wrote, the main technique used is X-ray diffraction analysis, and IR plays the role of linking results for crystals and solutions. There are no other methods, including NMR, to perform such a study. Surprisingly, the Reviewer considers X-ray diffraction analysis to be insufficient for determining the structure of compounds, and they need to be confirmed by NMR for solutions.

The results of this study are necessary to start work on the use of the NMR method for the study of vinyl cations.

For all solutions, the IR spectra of which were used in the article, the NMR spectra were also recorded. They cannot be used to confirm the results of work for the following reasons:

  1. The NMR spectra are complicated by the fact that they contain signals of cations in different isomeric states, however, in the IR spectra, the frequencies of their С=С vibrations actually coincide. Therefore, the IR method is convenient to use in this work.
  2. Salts of vinyl cations are strongly associated in solutions. Therefore, the NMR spectra show both narrow signals of monomers (weak) and broad signals of associates of various degrees of association (strong). In the solutions with which we work, self-association is high and their NMR spectra show wide intense signals (their width is tens and even hundreds of Hz). NMR experts who have been shown these spectra say that they belong to polymers and cannot be immediately interpreted. Self-association has almost no effect on IR spectra.
  3. Vinyl cations are rather strongly solvated by solvent molecules, which affects the spectra. Moreover, the degree of solvation decreases with an increase in self-association, which also affects the signals parameters. Solvation does not affect the characteristic frequencies of IR spectra.
  4. In the NMR spectra obtained by us, the chemical shifts of the 1H/13C signals from the atoms of the C=C+ and СºÐ¡+ groups, on which the “+” charge is predominantly concentrated, are in poor agreement with the calculated ones. This requires a special study.

All this means that it is impossible to use the NMR spectra of individual solutions to identify the content of vinyl cations in them at the current level of our knowledge.

It is necessary to study the concentration dependences of NMR spectra and the influence of the nature of solvents. This requires a separate special study, which is beyond the scope of this work.

In our work, we have shown ways to start such research. The reviewer's demand that we conduct such a study now and publish it in this article is perplexing. The reviewer is not familiar with the features of vinyl cations.

In conclusion, the findings cannot be published in a reputable scientific journal without NMR. In regard to the X-ray characterization that appears in a past publication, it is known that X-ray is not an ideal method for locating the protons

In the present work, knowledge of the exact localization of protons is not required. It is enough to define all groups С-H, CH2, CH3 and C=C bonds and this is successfully done.

Other minor issues are that the structure of the paper is not proper, lacking an introduction, experimental, results and discussion section and finally a conclusion. It rather reads as an essay of theories that are only IR supported and in my opinion are wrong.

This work is Communication and is framed according to the rules for Communication. The referee should be familiar with these rules.

Specifically, the authors rely specifically on IR spectroscopy for the characterization of the again very interesting chemical species.

I wrote above that the referee's opinion on how the IR spectroscopy method is used in the work is not correct. The Referee's requirement to prove by NMR that vinyl cations are formed in solutions, regardless of the results of X-ray diffraction analysis, is the wrong way and is not suitable for this work.

Round 2

Reviewer 2 Report

I wrote that the authors, in this particularly study, rely only on IR spectroscopy and this is completely true. The Editors can check whether this is true or not. I don't see any crystal reported on this paper. I suggest to them to obtain NMRs or if you have obtain them, to make them available to the reviewers.